# Evolution of Wearable Devices with Real-Time Disease Monitoring for Personalized Healthcare

**DOI:** 10.3390/nano9060813

**Published:** 2019-05-29

**Authors:** Kyeonghye Guk, Gaon Han, Jaewoo Lim, Keunwon Jeong, Taejoon Kang, Eun-Kyung Lim, Juyeon Jung

**Affiliations:** 1BioNano Technology Research Center, Korea Research Institute of Bioscience and Biotechnology (KRIBB), 125 Gwahak-Ro, Yuseong-Gu, Daejeon 34141, Korea; will1024@kribb.re.kr (K.G.); 7chawon@kribb.re.kr (G.H.); zeuyim5052@kribb.re.kr (J.L.); kwjeong@kribb.re.kr (K.J.); kangtaejoon@kribb.re.kr (T.K.); 2Department of Nanobiotechnology, KRIBB School of Biotechnology, University of Science and Technology (UST), 217 Gajeong-Ro, Yuseong-Gu, Daejeon 34113, Korea

**Keywords:** wearable devices, real-time monitoring, personal health, biosensor, portable devices, attachable devices, implantable devices, personal health, point-of-care, physiological signals

## Abstract

Wearable devices are becoming widespread in a wide range of applications, from healthcare to biomedical monitoring systems, which enable continuous measurement of critical biomarkers for medical diagnostics, physiological health monitoring and evaluation. Especially as the elderly population grows globally, various chronic and acute diseases become increasingly important, and the medical industry is changing dramatically due to the need for point-of-care (POC) diagnosis and real-time monitoring of long-term health conditions. Wearable devices have evolved gradually in the form of accessories, integrated clothing, body attachments and body inserts. Over the past few decades, the tremendous development of electronics, biocompatible materials and nanomaterials has resulted in the development of implantable devices that enable the diagnosis and prognosis through small sensors and biomedical devices, and greatly improve the quality and efficacy of medical services. This article summarizes the wearable devices that have been developed to date, and provides a review of their clinical applications. We will also discuss the technical barriers and challenges in the development of wearable devices, and discuss future prospects on wearable biosensors for prevention, personalized medicine and real-time health monitoring.

## 1. Introduction

Today, with the increase in the elderly population, the medical industry has changed dramatically, with a focus on the development of biosensors that enable real-time health monitoring, prevention and personalized medicine for a variety of chronic and acute diseases. Point-of-care technology (POCT) provides rapid and patient-centered diagnostics, especially for those with limited access to health services, while conventional disease diagnostic tests commonly used in laboratories and hospitals are time-consuming and costly, and require highly trained personnel. As healthcare regimes shift more toward personalized medicine, globally wearable sensors will have an average compound annual growth rate (CAGR) of approximately 38% from 2017 to 2025, among which the development of the smart watch is expected to grow at a particularly high rate [1].

Wearable biosensors have received tremendous attention over the past decade, mainly concentrated in the healthcare industry, which attempts to apply physical signals such as heart rate, blood pressure, skin temperature, respiratory rate and body motion to extract clinically relevant information [2,3,4,5]. Wearable devices are real-time and noninvasive biosensors allowing for the continuous monitoring of individuals, and thus provide sufficient information for determining health status, and even preliminary medical diagnosis. In addition, wearable biosensors allow health care providers to monitor the physiological traits of patients after therapeutics or treatments.

Wearable biosensors refer to biological sensors, including watches, clothing, bandages, glasses, contact lenses and rings, that are conveniently attached to a person’s body, and provide a function that distinguishes them from existing devices in terms of portability, ease of use and environmental adaptability [6,7,8]. Wearable devices have gradually been developed in the form of accessories, integrated clothing, body attachments and body insertions (Figure 1A) [9,10]. Over the past few decades, the tremendous advance in electronic, biocompatible materials and nanomaterials has led to the development of implantable devices that enable diagnosis and prognosis through small sensors and biomedical devices, greatly improving the quality and efficacy of health care. Despite motion artifacts, one of the major problems associated with wearable sensors is the development of stretchable and skin-attachable electronic devices that can continuously and unobtrusively monitor human activity and vital signs without any disruption or restriction by the user’s movement. The first implantable medical device developed was a pacemaker for arrhythmia patients in 1958 [11]. Since then, various types of pacemakers and implantable cerebellar stimulators have been developed and used. In recent years, flexible and stretchable electronic devices have allowed implantable systems to be deployed in the deep brain, the intravascular area, the intracardiac area and even the single-cell interior [12,13]. At present, wearable devices are driven by their own receiver, feature a signal processor, and are battery-powered, enabling them to operate as a “microcomputer” and allowing for the connection of all processes, from information collection and processing, to communication and power supply. Wearable devices connect to other smart devices via Bluetooth, infrared, radio-frequency identification (RFID) and near-field communication (NFC) technology. Together, this connectivity has led to the development of wearable systems for remote and long-term patient monitoring in homes and communities that were previously impossible (Figure 1B). This capability is expected to make a significant contribution to reducing medical and healthcare costs in countries with a large population of elderly people. This article provides a review of the evolutionary clinical applications of commercially available, newly emerging, technically challenging and future wearable devices. We also discuss the technical barriers and challenges of currently available biosensors and the future prospects for emerging biosensors.

## 2. Portable Devices

From the early 21st century, wearable devices have begun to provide personalized health services, as well as advanced levels of personalized portable devices and sensors. Currently, portable devices can be classified as wrists (watches, bracelets and gloves), heads (glasses and helmets), body clothes (coats, underwear and pants), feet, and body sensory control devices (somatosensory modulators) (Figure 2, Table 1) [2,8,14,15,16]. Due to the miniaturization of biosensing devices, and the development of microelectronics and wireless communication technologies, the wearable monitoring systems deeply embedded in our lives have been further developed [17].

### 2.1. Wrist-Mounted Devices

Wrist-mount devices for physiological monitoring have been developed commercially with improvement in battery longevity and miniaturization of hardware for converting raw signals to real-time interpretable data. Wrist-mounted devices, such as fitness bands and smart watches, are moving from basic accelerometer-based “smart pedometers” to include biometric sensing. Typical noninvasive monitoring devices carry out two functions: (1) Communication with electronic devices and (2) monitoring of human physiological signals and human activity signals [15,18].

Blood pressure measurement is one of the most important physiological indicators of an individual’s health status [19]. Conventional pulse wave sensors used cuffs to non-invasively monitor blood pressure and included optical, pressure, and electrocardiogram (ECG) sensors. However, these sensors are large in size, difficult to handle, and cannot be accurately measured when the subject moves during blood pressure measurement. To solve this problem, Lee’s group developed a wearable device with a Hall device that can detect the minute changes of the magnetic field of the permanent magnet and obtain the pulse wave data. This device can be worn on the wrist, and is a pulsimeter without a cuff [20]. Hsu et al. presented a prototype skin-surface-coupled personal wearable health monitoring system that captures high-fidelity blood pressure waveforms in real time and communicates with wireless devices such as smart phones and laptops [21].

Recently, various applications using a photoelectron imaging (PPG)-based heart rate sensor mounted on the wrist have been proposed [22]. The bracelet-type PPG heart rate sensor developed by Ishikawa et al. detects changes in heart rate and shows the possibility of overcoming motion artifacts in daily activities. Calibration of noise-free heart rate detection was measured using noise reduction pulse signals based on peak detection and autocorrelation methods [15,23].

Smart watches are one of the most popular wearable device types, and GlucoWatch^®^ biographer (Cygnus Inc., Redwood City, CA, USA) is the first to have a commercially approved non-invasive glucose monitor by the Food and Drug Administration (FDA) [24]. It electrochemically acquires information about glucose concentration extracted by reverse iontophoresis from skin interstitial fluid. Glennon et al. introduced a watch including fluid systems and storage systems, which can monitor sodium content in the body from sweat in real time [25]. In addition, the wrist-mounted device is applied in the measurement of daily activity including motion, gesture, rotation, acceleration and patient monitoring.

For monitoring of Parkinson’s disease (PD) patients, the smart watches can be used to analyze tremor and balance dysfunction with a gyroscope or accelerometer [26]. Roberto’s group assessed smart watches for quantification of tremor in PD patients, evaluation of clinical correlation, and its acceptance and reliability as a monitoring instrument. As a result, the smart watch has the possibility as a clinical tool and good acceptance by patients. In addition, Tison’s group used the smart devices for developing an algorithm to detect atrial fibrillation (AF) from the data of heart rate measured with PPG sensor and step count with the accelerometer [27]. The main cause of stroke is AF, and patients at risk of stroke can prepare for the disease by continuously monitoring AF.

### 2.2. Head-Mounted Devices

Wearable smart glasses are a type of head-mounted computer that displays information [28]. Smart glasses designed by Nicholas Constant et al. are pulse-sensing glasses containing a photoplethysmography (PPG) sensor on the nose pad to continuously monitor heart rate [29]. Joseph Wang et al. presented eyeglasses comprising nose pads consisting of a lactate biosensor capable of monitoring lactate, as well as a potassium ion-selective electrode that can measure potassium ions from sweat in real time [30]. Smart glasses can also be fabricated with sensors such as accelerometers, gyroscopes, altimeters, barometers, magnetometers and GPSs. The more advanced smart glasses are called Recon Jet, and they aim to capture information about their owner’s health status while running or riding a bicycle, by providing information about their activities through the display [31]. Mitsubayashi et al. produced a mouthguard glucose sensor using microelectromechanical systems (MEMS), and includes Ag/AgCl and Pt electrodes fabricated with enzyme membrane immobilized glucose oxidase [32]. Kim et al. demonstrated an enzyme-based biosensor integrated mouthguard for detecting salivary uric and lactate [33,34]. The enzyme modified system demonstrates high selectivity and sensitivity.

### 2.3. E-Textiles

Smart textiles, or smart clothing, consists of conductive devices and clothing material that is attached to or woven with the conductive devices. Textile-based diagnostic devices incorporate sensors, such as electrodes and fabrics, or by sewing electrodes into fabric [35,36]. The integrated sensors into textiles have been used to analyze biofluids [36]. Smart textiles must include three components. i.e., a sensor, an actuator and a controlling unit [37]. Textile containing electrodes, called e-textiles, are used to monitor human physiological signals, biomechanics and physical activity, such as motion, body acceleration and pressure [18,38].

To measure glucose and lactate with high accuracy, Liu et al. fabricated a glucose and lactate detection system by using glucose oxidase and lactate oxidase-based electrodes in a fabric [36]. In addition, Liu’s group developed living material and a glove that has integrated hydrogel-elastomer hybrids integrated with genetically engineered bacteria including genetic circuits, to give materials desirable functions [39]. Chemically different induced cell strains were encapsulated in the chamber of hydrogel, and interaction between the bacteria strains and environment is created via diffusion.

When inducer (IPTG, Rham) is contacted with the bacterial sensor that is programmed fluorescence (IPTGRCV/GFP, RhamRCV/GFP), the fluorescence response is activated. The biosensor with synthetic biology technology has the potential to monitor healthcare and environment as mechanical flexibility and low cost.

Physiological signals, such as heart rate, breathing rate and temperature, are also monitored using e-textiles [40,41]. The Hexoskin wearable vest is capable of monitoring heart rate and breathing rate during daily activity [42]. An electronic shoe has been developed to measure walking ability, such as lateral plantar pressure, heel strike, toe pressure, and ground reaction forces [43,44], which provide fundamental information for distinguishing among gait phases. Rupesh K. Mishra et al. developed a glove incorporating an electrochemical biosensor with a stretchable printable enzyme-based electrode that is able to detect organophosphate (OP) nerve-agent compounds. The glove consists of an index finger that has a carbon-based counter electrode, working electrodes, an Ag/AgCl-based reference electrode, and a thumb printed carbon pad. The index finger is a sensing finger that contains an organophosphorus hydrolase layer, and the thumb is a collector/sampling finger. Furthermore, stress-enduring inks are used to print the electrode system and the long serpentine connections to the wireless electronic interface. This lab-on-a-glove is applied as a point-of-use screening tool and in defense and food security applications [45].

### 2.4. Others

Smart jewelry, a wearable real-time monitoring device, is designed to warn users through smartphone alerts whilst tracking smart biomedical signals and biomechanics at the time of payment or performing out ambient sensing [18]. In addition, these wearable devices can track human activity, such as consumed calories and distance travel, and measure quality and duration of sleep [46,47,48]. Bellabeat Leaf is a smart accessory, such as a necklace, a bracelet, or clip that uses only 3D accelerometers and haptic vibration motors that detects sleep, daily activities, menstrual cycles, and breathing, and do not have commonly used sensors like GPS or a heart rate monitor [49]. The smart belt such as WELT (Wellness Belt) or BELTY monitor information such as waist size, food intake and movement of users like step count, and sitting time [50,51,52]. As a smart chest strap, OmegaWave has monitoring electrodes providing information of electrocardiogram (ECG) and direct current for assessing the activity of the cardiac system and central nervous system [53]. The Zephyr Bioharness is another chest strap that features reliable real time recording of HR, breathing rate, temperature, ECG and respiratory rate [54]. In particular, respiratory rate is detected by measuring chest expansion and contraction [55].

## 3. Attachable Devices

Attachable monitoring devices are considered to be the next-generation personal portable health care devices for remote medical progression. An important feature that defines an attachable device is skin-like adaptability and flexibility, providing accurate and reliable sensing without compromising a user’s natural movement and comfort. Flexible thermoplastic polymers such as PC, polyethylene terephthalate (PET) and polyurethane have been selected for flexible materials production due to their excellent optical transparency, ease of manufacture and excellent deformability. In addition to soft substrate-based templates, completely functional, attachable and flexible sensors are essential for active sensing elements—their most important components [56].

With the latest advances in sensor technology, MEMS, microelectronics, data analysis, communications and physiotherapy, attachable devices can be developed. In particular, the miniaturization of electronic circuits using microelectronics has been an important part of the development of attachable devices. In the past, due to the size of sensors and front-end electronic devices, the hardware had difficulty in collecting physiological and biomedical data for long-term monitoring applications. Currently, the development of microelectronics allows the creation of microcontroller functions and wireless transmission-enabled circuits. MEMS also allowed miniaturization that enabled batch manufacturing, and significantly reduced the cost of electronic components [14]. Smart attachable sensing devices are an important component in real-time health monitoring systems for physiological signals that are closely associated to physical conditions such as blood pressure, heart rate, electrophysiology, body temperature and various sweating biomarkers.

### 3.1. Wearable Skin Patches

Wearable skin patches are becoming increasingly pervasive within the wearables market. Soft, flexible and stretchable electronic devices are connected to soft tissue to provide a new platform for robotic feedback and control, regenerative medicine and continuous healthcare [57]. Skin patches are ideal wearables, because they can be obscured by clothing, and can record more accurate data without being disturbed by movement. Wearable patches worn on the human skin have been utilized as cardiovascular, sweat, strain and temperature sensors [58].

#### 3.1.1. Monitoring of Blood Pressure and Heart Rate

Monitoring cardiovascular signals such as blood pressure and the heart rate of patients receiving medical care is very important. A thin, flexible and patch-type continuous blood pressure (BP) monitoring sensor is constructed with a layered structure of ferroelectric film, specially designed electrodes and flexible electronic circuits, which together enable simultaneous electrocardiogram (ECG) and ballistocardiogram (BCG) measurements on the human chest without discomfort [59]. In a feasibility study using the developed sensor, the estimates of systolic blood pressure are in good agreement with the reference value, and the correlation coefficient of the class was 0.95 (*p* < 0.01).

Reported in Advanced Functional Materials, a wearable patch sensor incorporating flexible piezoresistive sensor (FPS) and epidermal ECG sensors for cuffless blood pressure monitoring has been developed [60]. The system simultaneously measures epidermal pulse signals and the ECG and obtains bit-to-bit BP data in real time through the pulse transit time (PTT) method. To obtain a very stable surface pulse signal, a parametric model of the FPS detection mechanism was developed and the operating conditions were optimized. In particular, this sensing patch can operate at ultra-low power (3 nW) and detects subtle physiological changes such as before and after exercise to provide promising solutions for real-time and home-based BP monitoring.

In a recent study, Sheng Xu and colleagues demonstrated the initial function of a conformal ultrasonic patch to monitor blood pressure waveforms in areas of deep arteries and veins [61]. Ultrasonic waves can penetrate deep into biomechanical tissues, enabling 3D detection of currently worn electronic devices. 

Wearable ultrasonic devices ensure intimate, conformal contact with curved and time-dynamic skin surface, and allows continuous monitoring of CBPs for cardiovascular disease without operational discomfort or instability caused by other traditional methods (Figure 3). The device is built around piezoelectric ultrasound transducers with a 4 × 5 arrangement, and is connected to 20 stimulating electrodes respectively. The array is designed to map the position of the vessels so that it can be detected and monitored using a transformer above the target without the need for tedious manual placement.

The wearable pulse wave monitoring sensor is attached directly to the epidermis, and the fluctuations of the pulse wave can cause the transform of piezoelectric material of sensor (Table 2) [62,63]. D.Y. Park et al. present a self-powered, flexible piezoelectric pulse sensor based on high-quality lead zirconate titanate (PZT) thin film for a real-time arterial pulse monitoring system [64]. Experimental results demonstrate a piezoelectric pressure sensor with a sensitivity of 0.018 kPa^−1^, response time of 60 ms, and remarkable mechanical durability. The developed sensor connected with a wireless blue-tooth transmitter and an Android-based smart phone which could display the detected signals in real time.

Attachable and flexible pulse sensors are designed to detect long-term biological signals [65]. One device was incorporated with microthin inorganic photodetectors (IPDs, thickness: 4.1 μm) and a red light-emitting diode (620 nm), and is encapsulated into an adhesive elastomer layer. The device operates in a reflective mode and can be attached to various locations on the human body to measure the heart pulse waveforms. Using microstructured polydimethylsiloxane (PDMS) (i.e., pyramid-like patterns) films, Bao et al. introduced a continuous, real-time pressure monitoring system with wireless, flexible, passive and millimeter-scale sensors [66]. This monitoring system is used to capture human radial pulse waveforms in real time, as well as to continuously monitor in vivo intracranial pressure in proof-of-concept mice studies. In a recent study, a highly sensitive flexible three-axis tactile sensor is presented by combining micro-pyramid PDMS arrays and a reduced graphene oxide (rGO) film [67].

#### 3.1.2. Monitoring of Bodily Fluids

As a typical body fluid, sweat is especially important because it contains large amounts of important biomarkers, including electrolytes, small molecules, and proteins [68]. Over the past few years wearable sensors have been developed for sweat analysis and have detected various sweat components.

The ultrathin, flexible wireless sweat sensor is installed on functional elastomer substrates, enabling epidermal analysis of biocompatible fluid [69]. This sweat sensor measures the amount of sweat and its chemical properties through the detection of the dielectric and colorimetric. The device contains an inductive coil for stretch and flexibility, and planar capacitors with interdigitated electrodes. The device includes a functional soft substrate that can spontaneously collect sweat through capillary forces without operating any complex microfluidic handling system. Certain components (OH^−^, H^+^, Cu^+^ and Fe^2+^) of sweat can be detected in a colorimetric measurement mode by introducing indicator compounds deep in the substrate in the same system.

According to the experimental results, the amount of sweat volume of 0.06 μL/mm^2^ can be accurately measured with high stability and low drift, and color reactions to pH and various ions provide functions related to sweat analysis.

Continuous and non-invasive biomarker monitoring is important to manage human health, performance and well-being in the field of exercise science and medicine [70]. A wearable electronic sensor capable of simultaneously measuring lactate, hydrogen ions, and sodium ions in human sweat through temperature sensing for internal calibration, is equipped with microfluidic sampling and wireless reading electronics [71]. On the equipment platform, sweat passes through a flexible microneedle sensor array integrated into the microfluidic channel with continuous flow. A potentiometric sodium ion sensor has been developed by depositing a polyvinyl chloride membrane into the internal layer of electrochemical deposited poly (3,4-ethylenedioxythiophene) (PEDOT), and the pH detection layer is based on a high sensitivity of an iridium oxide (IrOx) membrane. The amperometric-based lactate sensor consists of doping enzymes deposited at the top of a semi-penetrating copolymer membrane and outer polyurethane layers, providing good selectivity when various analytes are present.

Tomczak et al. has developed a fully integrated, wearable, flexible and wireless sweat detection device that can measure hydration status continuously and non-invasively by monitoring electrolytes during intense exercise [72]. The main differentiating features of the developed device are as follows: (1) A conformal fluid system that effectively collects sweat from the skin with a high rate of sweat absorption and removes it quickly from the detection area to minimize effects on sweat physiology. (2) Flexible microfluidics and low noise footprint electronics that combine Na^+^ and K^+^ ion selective electrodes to enable wearable, wireless sweat monitoring.

Recently, an electrochemical sensor that can be attached to the skin to detect glucose and pH in sweat has been studied [73]. Patterned elastic electrodes are manufactured by layering a deposition of carbon nanotubes on a pattern of Au nanosheets filtered on a flexible substrate. The fabricated electrochemical sensors are highly sensitive and selective for both pH and glucose in sweat because of its close adhesion to the skin using the sticky polymer, Silbione, with mechanical stability corresponding to 30% stretching and air stability 10 days.

Wearable biosensors developed to monitor noninvasively and constantly target biomarkers, have measured single sample biofluids. Recently, researchers have been working hard to develop an epidermal biomonitoring system that measures two different body fluids (skin interstitial fluid and sweat) using a single wearable sensing platform. This dual specimen collection and epidermal sensing system has been realized through the parallel operation of reverse iontophoretic ISF extraction across the skin and iontophoretic delivery of a sweat-inducing drug (pilocarpine) into the skin at separate sites [74].

Because diabetes is a chronic condition caused by insulin control problems that can lead to various major complications, the patient’s glucose levels should be constantly monitored and controlled. Therefore, attachable devices that monitor glucose to improve diabetes management and blood glucose control is very attractive. PDMS dermal patch is a microfluidic sampling system combined with a thermal ablation system that allows noninstrusive control and sampling of glucose or other biomolecules present in interstitial fluids without invasive extraction [75]. Kim’s group has created a wearable patch-type sensor made from graphene and gold to enable sweat analysis based diabetes therapy [76]. The flexible devices used gold mesh and gold-doped chemical vapor deposition (CVD) graphene to ensure high conductivity, optical transparency and mechanical reliability for stable electrical signal transmission, and to give translucency such as skin in large deformable device arrays. GP-hybrid sensors use gold-doped CVD graphene as an electrochemically active soft material for enhanced electrochemical activity, selectivity and sensitivity to detect important biomarkers in human sweat. The patch consists of sweat control components (a sweat-uptake layer and waterproof film), sensing components (humidity, glucose, pH and tremor sensors) and therapeutic components (microneedles, a heater and a temperature sensor) that can be thermally activated to deliver drugs transcutaneously (Figure 4A).

When the relative humidity is detected at the critical sweating point, the temperature and pH is measured and corrected at the same time. When a high concentration of glucose is measured, the internal heater dissolves the phase-change material and releases Merformin in a feedback percutaneous drug delivery reaction through biocompatible microneedles.

Researchers at Binghamton University at New York State University have developed a new self-powered, disposable and wearable sensor patch that can monitor glucose in human sweat to detect hypoglycemia associated with exercise [77]. This glucose sensing biosensor incorporates a paper-based glucose/oxygen enzymatic fuel cell that is stacked vertically in a standard Band-Aid adhesive patch. The paper-based sensors are directly attached to the skin to monitor glucose without external power supply and advanced reading equipment. The calibration curve for the developed sensor with a 10 kΩ resistor represents a highly linear output signal at 0.02 to 1.0 mg/mL glucose (R^2^ = 0.989) with a high sensitivity of 1.35 μA/mM. 

In a more recent study, Heikenfeld’s lab reported complete validation of a blood-correlated sweat biosensing device with integrated sweat stimulation using carbachol as stimulant, microfluidic transport using a hexagonal wick, and an alcohol oxidase [78]. A fully integrated device capable of continuously measuring sweat ethanol, which is accurately correlated with blood ethanol. Eccrine systems, an advanced sweat sensor company, uses Heikenfeld’s invention to commercialize this wearable sweat-sensing device. A commercial alcohol biosensor analyses the sweat. The sensor measures concentration of hydrogen peroxide that is generated by metabolizing the ethanol with alcohol oxidase on the enzymatic electrode.

#### 3.1.3. Monitoring of Body Temperature

It is very important to measure changes in skin temperature during the initial diagnosis and treatment of the disease [80,81]. Rogers’s Group demonstrated ultra-thin skin type sensors that can be flexibly attached to the skin surface for continuous and accurate thermal characterization [82]. The ultra-thin and compliant structure of these devices offer significant benefits. First, the skin-like properties are firmly attached to the skin without irritation and are effectively isolated from the strain applied by the sensors/actuators. Second, the thermal mass of the device is extremely low, and its water/gas permeability is high, which is advantageous in terms of response time and thermal loading. The group also developed ultrathin photonic devices by combining colorimetric temperature indicators with wireless flexible electronics. The device used thermochromic liquid crystals formed of a large pixel array on a thin elastomeric substrate [83]. An algorithm that analyzes color patterns recorded on a device using a digital camera, and tools that correlate results with the fundamental heat treatment process of skin surfaces, make the resulting data useful. This epidermal photonic system has a tremendous ability to characterize the skin, and provides important parameters for determining physiological conditions and cardiovascular health.

Reported in Advanced Materials, transparent and stretchable (TS) sensors that simultaneously monitor subtle changes in skin temperature and deformation during human activity are made into a simple process, which can easily be attached as a patch to an object or to the body [79]. The TS-gated and TS-resistive temperature sensing devices exhibited a high sensitivity of approximately 1.34% per °C, and there was no change in response after 1000 cycles of stretching at 30% strain.

In order to reduce the negative effects of psychological stress upon human society and health, psychological stress should be constantly monitored in daily life. Therefore, researchers have developed a flexible human stress monitoring patch that reduces skin contact area and improves patch wear (Figure 4B) [84]. The human stress monitoring patch is created by integrating three sensors that can track skin temperature, skin conductance, and pulsewaves with stamp size (25 mm × 15 mm × 72 mm). The development of integrated multi-layered structures and associated micro-manufacturing processes minimizes skin contact areas, resulting in a reduction of 1/125 of the traditional single-layer multiple sensors. The flexibility of the patch has been increased by the invention of a flexible pulse waveform sensor made from flexible piezoelectric membranes supported by perforated polyimide membranes with high chemical resistance and flexibility. The assembled patches measured the skin temperature with a sensitivity of 0.31 Ω/°C for the human physiological range, and the pulse response time was 70 ms.

### 3.2. Contact Lens

Smart contact lenses can monitor the physiological information of the eyes and of any tears non-invasively. Several types of contact lenses using optical and electrical methods have been developed to monitor the chemicals (lactate and glucose) and electrical conductivity of the tear fluid and the transcutaneous gases in the eye’s mucous membrane. Alexeev et al. has developed photonic crystals composed of a face-centered cubic arrangement of colloidal particles embedded in hydrogel applied to non-invasive glucose sensing of tear fluid [85]. For glucose sensing at physiologic pH values, new boronic acid derivatives, such as 4-amino-3-fluorophenylboronic acid (AFBA) and 4-carboxy-3-fluorophenylboronic acid (CFBA), have been used as a molecular recognition agent that combines with glucose to form bis-bidentate cross-links.

In response to the glucose, hydrogel cross linking increases elasticity recoverability, constricts the hydrogel volume, and causes diffraction to blue shift in proportion to the glucose concentration. The color changes that are visualized without any measurement are changed from red to blue in the visible spectrum, depending on the physiological glucose concentration.

A fluorescent contact lens with a hand-held photofluorometer was proposed for the noninvasive monitoring of glucose [86]. The contact lenses were made from liquid hydrogel nanospheres containing tetramethylrhodamine isothiocyanate concanavalin A (TRITC-Con) and fluorescein isothiocyanate dextran (FITC-dextran). As the glucose concentration increases, glucose displaces FITC-dextran from the combined position on TRITC-Con A, thereby increasing the fluorescent intensity.

A wearable contact lens optical sensor has been developed to continuously measure glucose in physiological conditions [87]. The sensor is fabricated on the surface of a glucose-sensitive hydrogel network using a simple stamping method, and is attached to commercial contact lenses. Using smartphone applications, the sensor can record the reflectivity of the primary diffraction light, and it demonstrates the benefits of high sensitivity, fast response time and short saturation time (Figure 5).

The electronic enzyme L-lactate sensor, using contact lenses, is designed to detect L-lactate in the tear fluid potentially at least invasive [88]. The sensor utilizes a functional platinum structure with cross-linking of lactate oxidase, glutaraldehyde and bovine serum albumin, and is coated with medical polyurethane. The sensor can measure the physiological concentration of L lactate on the tear film based on its fast response time of 35 s, an average sensitivity of ~53 μAmM^−1^cm^−2^ within a linear range, and sufficient resolution.

## 4. Implantable and Ingestible Devices

Implantable or implanted devices have recently become an emerging measurement of wireless medical measurement (Table 3). This measurement is possible through the fusion and development of MEMS technology with biology, chemistry, electrical and mechanical technology. Since these devices operate directly in the body, they should be prevented in advance from adverse effects (such as rejection of transplantation) that may affect the human body. Because human tissue is conductive, it can short-circuit the antenna of the device if it is in direct contact with the metal material of the implanted or ingested device [89]. Insertion-type instruments are used to diagnose and treat diseases by detecting changes in the body, and ingested instruments are considered to be suitable for endoscopy because they pass through the digestive system. The wireless remote capability of these devices is essential, not only for transferring patient monitoring data, but also for maintaining the device’s battery and status and function upgrades. So researchers have been trying to solve the problems of implantable or consumable devices that will be used in the future (antenna design and performance enhancement, configurable radio frequency settings, power and electronic system modeling).

### 4.1. Implantable Devices

Since cardiac pacemakers were first developed in the 1960s, the number of patients treated with cardiovascular implantable electronic devices, including pacemakers, implantable cardiovascular defibrillators (ICDs) and implantable deep brain stimulators, has been increasing [92,95]. Most implantable devices consist of batteries and biocompatible materials, as well as programmable circuits. The pacemaker is the most well-known implanted medical device for heart patients; the device is used to treat irregular heartbeats known as arrhythmias, and provides low-energy electrical pulses to restore normal rhythm when irregular heartbeats are detected. The ICD is the latest version of the pacemaker that operates in the same manner. Sudden cardiac death (SCD) accounts for half of all deaths from heart disease. If a conventional pacemaker is not able to restore normal rhythm to heart rate, an ICD will provide a high-energy electrical pulse; indeed, the ICD is associated with significant reductions in mortality among patients at high risk of SCD from ventricular arrhythmias. Deep brain stimulation has been introduced as an effective means of treating movement disorders such as Parkinson’s disease [93]. The procedures for transplanting deep brain stimulation electrodes require stereotactic surgery aimed at the neurological structure. The targeted electrodes are controlled by an implantable pulse generator (IPG) for deep brain stimulation to provide electrical signals to control movement. The IPG consists of a battery that generates an electrical stimulus and regulates the electronic circuitry, and thus supplies energy to the target neural system [94].

Tattoos are fascinating platforms for monitoring emotions and vital signs. Electronic tattoos (e-tattoos) can be adapted to various skin textures, enabling noninvasive and best attachment methods to the skin. The texture of the tattoo adhesive layer is thoroughly flexible to move with any skin movement, providing natural wearability for the patient and precise data to the physician. Currently, electronic tattoos function as a means of diagnostic and monitoring for primary healthcare providers who want optimum clinical decisions.

MaApline et al. has developed wireless graphene nanosensors that do not require a power source, using biomaterials such as tooth enamel to remotely detect and monitor bacteria present in breathing or saliva [102]. The advantages of graphene nanosensors are biocompatibility, robustness, optical transparency, biotransferability and flexibility. Because of these features, graphene printing on a water-soluble silk film substrate can serve as a temporary tattoo platform. Graphene nanosensors using antimicrobial peptides have the ability to specifically detect pathogenic bacteria at a single cell level. The Wang group has recently developed a non-invasive test method based on tattooing to monitor the presence of lactate, glucose, ammonia and alcohol in the body. Tattoo biosensors, which measure the concentration of lactate in human sweat in a noninvasive manner, have been developed to monitor electrochemical signals generated by enzymes [96]. A new skin biosensor functionalized with lactic acid oxidase showed linear high specificity up to 20 mM for lactate secreted from the sweat glands. In addition, the tattoo sensor has a flexible feature, so it has a strong durability even when the skin moves repeatedly. In fact, sensors have been applied to analyze the change of lactate in real-time in the sweat glands of subjects with long-term repeated exercise. In contrast, an ammonia potentiometric tattoo sensor uses an ammonium selective polymer membrane, which is based on nonactin ionophores and a solid state reference electrode [97]. Physiological tests using a tattoo biosensor having an ammonium selective polymer membrane showed NH4^+^ values of 0.1 to 1 mM. The authors also succeeded in applying a tattoo-based non-invasive device platform to glucose monitoring, which is easy to wear and move with the skin [98]. The tattoo-based blood sugar detection system is formed by the combination of reverse osmotic pressure-derived epileptic glucose, and an enzyme-based current measurement biosensor, and an oxidized enzyme Prussian blue converter is used (Figure 6A). This sensor responds sensitively to a 23 nA/μM glucose concentration, and has a specific response up to 3 μM. The validation method was performed by attaching a sensor to the subject’s skin and detecting changes in blood glucose after eating. The results of this study suggest that tattooing platforms using iontophoresis and biosensing may be effective in diabetes management and non-invasive monitoring using materials other than glucose in the interstitial fluid. Similarly, a wearable iontophoretic-biosensing temporary tattoo system was used to analyze real-time alcohol content from sweat [99]. The tattoo-type biosensor firstly transfers the drug called phyllocarpine to the transdermal preparation to generate sweat, and secondly measures the alcohol in real time using the alcohol-oxidizing enzyme built into the sensor. This biosensor was designed to be very suitable for the human body, and the experimental results showed a clear difference between before and after drinking. The alcohol sensor includes an electronic plate made of a flexible material in a wearable device for wirelessly measuring and controlling data.

Currently, most commercial wearable devices are using straps or adhesive tapes to attach the devices on the human body. The insecure skin-sensor interference makes these devices suffer from limited functionality and signal-to-noise ratio, as well as significant motion artifacts. In order to overcome these problems, researchers developed a low-cost, ultra-thin, tape-free and multifunctional electronic tattoo [106]. The sensor is manufactured by the cut-and-paste method, and has the filament structure, so it has excellent ventilation, and is not inconvenient to wear (Figure 6B). Since the electronic tattoo sensor is thin with a thickness of 1.5 μm, it can minimize the artificial feeling of moving with the skin. In addition, the electronic tattoo is equipped with various sensors, ECG (ECG) and skin temperature and moisture can be measured at the same time without any signal degradation.

Recently, researchers at Harvard and MIT have developed biosensitive inks (bioinks) that operate using a simple chemical reaction that does not require power for data processing and transmission, and have proven that the body surface itself can be used as a biointerface, namely, an interactive display [102]. Traditional tattoo inks have been replaced by color changing biosensors as the fluid of the interstitial fluid known as tissue fluid changes. The fluid interacts closely with plasma, so the fluid is a good indicator of the concentration of chemicals in the blood at any given time. The researchers examined four biosensors whose inks change color on the skin to monitor pH, glucose and sodium levels. To date, two types of inks have been tested in vitro using pig skin. One is designed to monitor the patient’s blood glucose levels, and changes from green to brown as the concentration increases. The second type turns light green when an increase in sodium is detected, and chases sodium concentration to prevent dehydration. In another study, Zhao et al. have developed a 3D bio-printing hydrogel ink that can print a programmed bacterial cell with a high resolution of about 30 μm on a large 3cm-sized biomaterial [101]. 3D printing inks contain waterborne, programmed bacterial cells, nutrients, and signal chemicals, with a blend of polymeric micelles and photoinitiators. In order to directly perform 3D printing on a biomaterial, first, a multi-hydrogel ink composed of various kinds of cells or chemicals (Figure 6C) is used to print, and second, ultraviolet rays are irradiated on the printed matter. The engineered bacteria cells can see the new features of the device through 3D printing. Live tattoos are printed on an elastomeric sheet consisting of a double layer of cells (responsive to AHL, Rham or IPTG) that sense various chemicals and are attached to the skin to incorporate various reactions (Figure 6C). Attached tattoos indicate green fluorescence in the corresponding 3D print pattern upon receipt of the chemical.

In some cases, implantable devices are no longer managed after a period of treatment. In this case, surgical retrieval procedures are often required, which imposes physical, biological and economical loads on the patient [107,108]. Hence, significant efforts from several research groups have afforded bioresorbable electronic implants for elimination of the devices after a certain period of time inside the body.

Recent studies have demonstrated a new class of multifunctional bioresorbable device with built-in optical elements [109], silk-based fully degradable therapeutic devices [110], bioresorbable electronic stent [111], and bioresorbable silicon sensors for the brain [112].

### 4.2. Ingestible Pills

A safe, non-invasive approach to accessing the fluid you want to identify is to use an ingestible sensor. This ingestible sensor is able to pass through the lumen of the digestive tract and reach organs around the abdomen. Thus, the ingestion sensor monitors the intrinsic genital contents and lumenal fluid as well as enzymes, hormones, electrolytes, microbial communities and metabolites around the organs and delivers biometric information [113].

Ingestible pills, introduced by Proteus Digital Health (Redwood City, CA, USA), are smart pills for monitoring the precise time at which any drug is taken. When the smart pill reaches the stomach, it is powered up by a chemical reaction with the stomach fluid and sends an ingestion time signal to the patch worn on the body. This patch not only communicates with the smart pill, but also monitors heart rate, blood pressure, pH and temperature (Figure 7) [103,105]. Currently, the most commonly used measure of compliance is indirect measurement through ingestion. My/Treatment/Medication (MyTMed) is a system that can directly check the compliance of drugs [104]. MyTMed consists of an electronic pellet that emits a radio frequency when it comes into contact with an acidic pH 2, and the Hub that receives the radio frequency and sends it to the cloud server, all of this allowing the patient and the doctor to communicate in both directions. The device can be controlled in real time with respect to drug intake information and compliance. Applying the MyTMed system to tablets will make it easier to manage many kinds of chronic diseases.

## 5. Conclusions

Wearable devices are becoming popular in various fields, from healthcare to biomedical monitoring systems (Table 4). In particular, wearable devices are becoming important for long-term health monitoring due to the increasing elderly population throughout the world [114]. In this paper, we have reviewed the latest advances in wearable sensor technologies to identify important biomarkers for noninvasive and possibly continuous monitoring of key diagnostic indicators. Certain technical challenges still need to be addressed for the wide-scale use and deployment of wearable devices as part of the digital health era. One such technical challenge is the personal calibration of wearable devices. The symptoms for early disease diagnosis may differ for every person because every person is unique, and various factors can affect personal health (e.g., family medical history, genetics and diet). Therefore, using a wearable device to monitor a patient’s health more accurately and properly requires a machine-based analysis of personal data and a personal calibration of the device. Another challenge is the misalignment of wearables, which affects the quality and accuracy of the measurements. Smarter designs are needed to tolerate such misalignments, for example, by using computational approaches and guide star-like internal references or self-calibration protocols. It is especially important to consider the variations in human physiology and the size or 3D conformation of different organs on which wearables operate [115].

As the sophistication and miniaturization in field of sensors, battery solution and material science have evolved, wearable biosensors have made many advances in terms of durability and robustness, and further improvements are required in the future. Wearable devices must be able to operate under a variety of conditions, such as in humid or wet environments or at warm temperatures, so that continuous parameter monitoring is possible without compromising performance. Another crucial wearable component that requires increased robustness is the battery, which is particularly important in GPS tracking, consuming a significant amount of battery power. Finally, next-generation wearables will be even smaller than current versions and as their size decreases, the packaging of sensors would be carried out in an integrated manner, so that they do not lose their efficacy in a small area, and can offer slim and even lighter-weight wearable designs.

Over the past few decades, the tremendous advancement of electronics, biocompatible materials and nanomaterials has led to the development of wearable devices that enable the diagnosis and prognosis of small sensors and biomedical devices, greatly improving the quality and efficiency of healthcare services. Future patient monitoring and clinical care will be based on efficient and affordable solutions for wearable devices, enabling remote and long-term patient monitoring in homes and communities that were previously impossible. Wearable devices are expected to contribute significantly to the elderly population’s health care and medical costs and to the development of personalized medical care.

## Figures and Tables

**Figure 1 nanomaterials-09-00813-f001:**
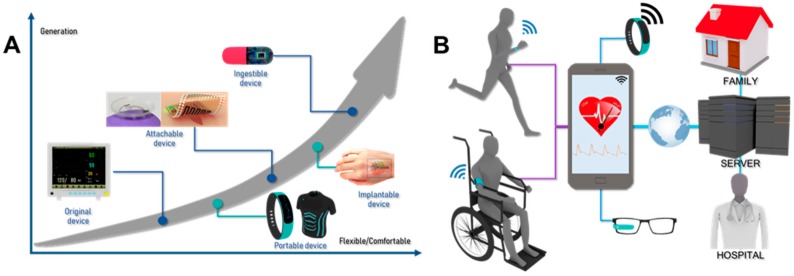
Industrial wearable technologies. (**A**) Evolution of wearable medical devices (**B**) Application of wearable devices in the healthcare and biomedical monitoring systems. Reproduced with permission from Hwang, I.; et al. Multifunctional smart skin adhesive patches for advanced health care; Wiley, 2018 [9] and Yao, H.; et al. A contact lens with embedded sensor for monitoring tear glucose level; Elsevier, 2011 [10].

**Figure 2 nanomaterials-09-00813-f002:**
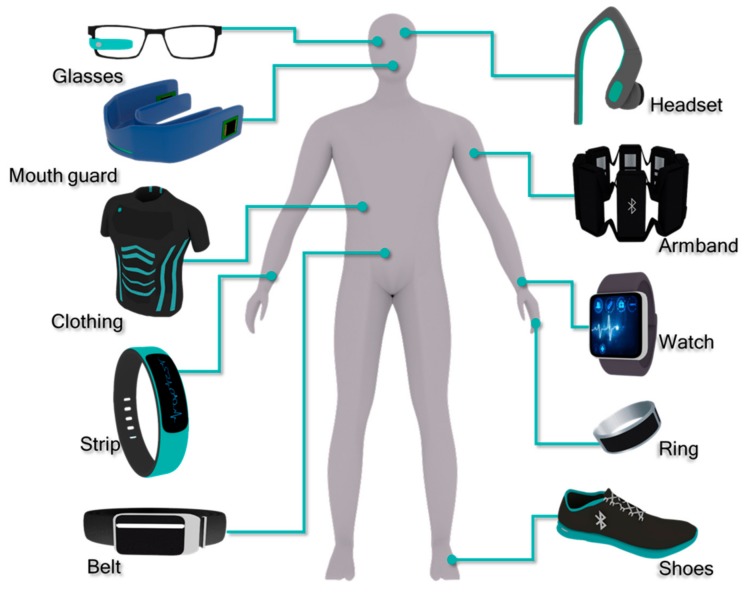
Portable medical and healthcare devices worn on body parts.

**Figure 3 nanomaterials-09-00813-f003:**
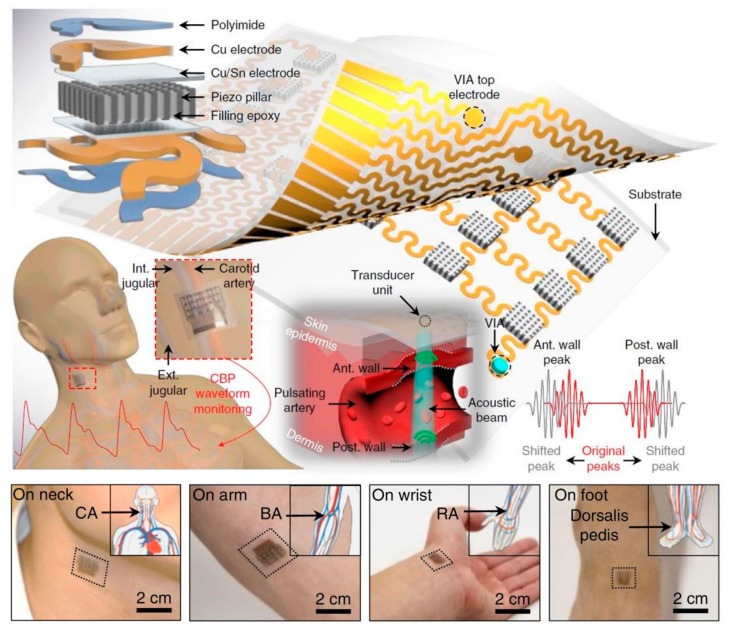
Stretchable ultrasonic device to identify and capture arterial blood-pressure waveforms. The high-performance 1–3 composite with piezoelectric microrods embedded in an epoxy matrix suppresses shear vibration modes and improves ultrasonic penetration into the skin. The wearable device can monitor peripheral vessel hemodynamics at different locations, for example, to sites of the brachial, radial, femoral or dorsalis pedis arteries. Reproduced with permission from Wang, C.; et al. Monitoring of the central blood pressure waveform via a conformal ultrasonic device; Springer Nature, 2018 [61].

**Figure 4 nanomaterials-09-00813-f004:**
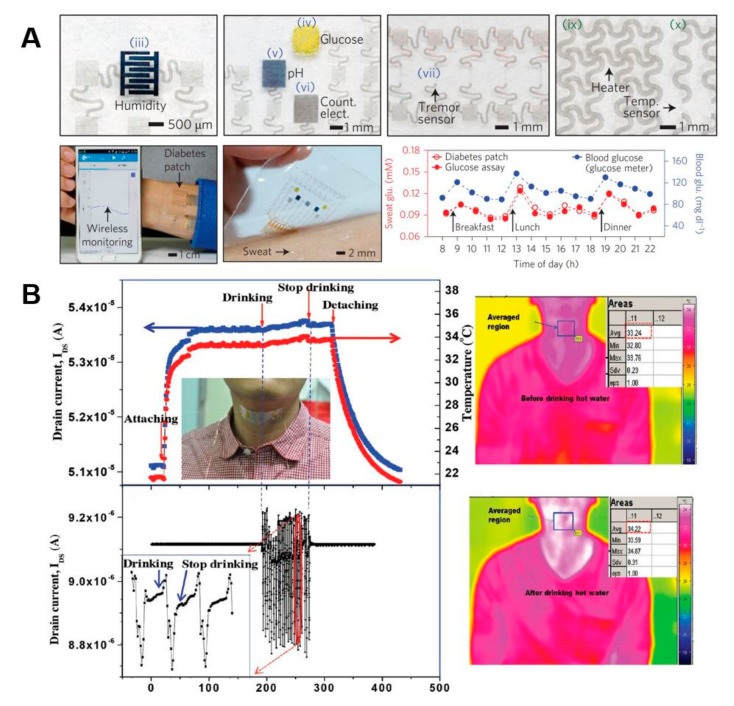
Flexible and stretchable sensing platforms. (**A**) Optical images of patch consisting of a humidity, glucose, pH, tremor, heater and temperature sensor. The diabetes patch is laminated on the human skin and is connected to a portable electrochemical analyzer with external devices via Bluetooth. Sweat glucose concentrations measured by the diabetes patch (red circles) and a commercial glucose assay kit (red dots) are well matched. In addition, changes in the sweat glucose concentration are well correlated with those of the blood glucose concentration. (**B**) The transparent and stretchable integrated platform of temperature and strain sensors shows the simultaneous responses to temperature of human skin during muscle movements or drinking of hot water when the integrated sensor platform was placed on the neck of a male subject. Reproduced with permission from Lee, H.; et al. A graphene-based electrochemical device with thermoresponsive microneedles for diabetes monitoring and therapy; Springer Nature, 2016 [76], and Trung, T.Q.; et al. An all-elastomeric transparent and stretchable temperature sensor for body-attachable wearable electronics; Wiley, 2016 [79].

**Figure 5 nanomaterials-09-00813-f005:**
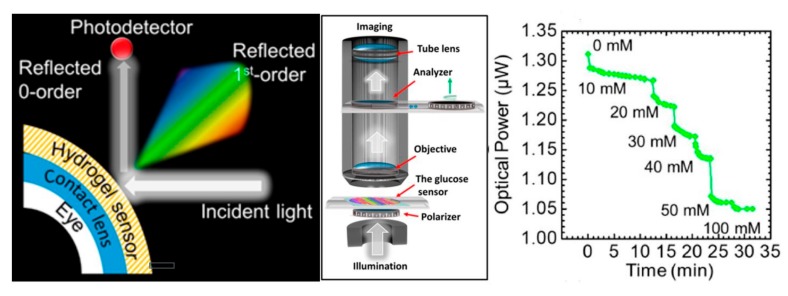
Examples of smart contact lens for detecting glucose. Schematic diagrams of the measurement setup. Continuous monitoring of the reflected power in response to various glucose concentrations (0–50 mM) versus time measured using the optical powermeter. Reproduced with permission from Elsherif, M.; et al. Wearable contact lens biosensors for continuous glucose monitoring using smartphones; American Chemical Society, 2018 [87].

**Figure 6 nanomaterials-09-00813-f006:**
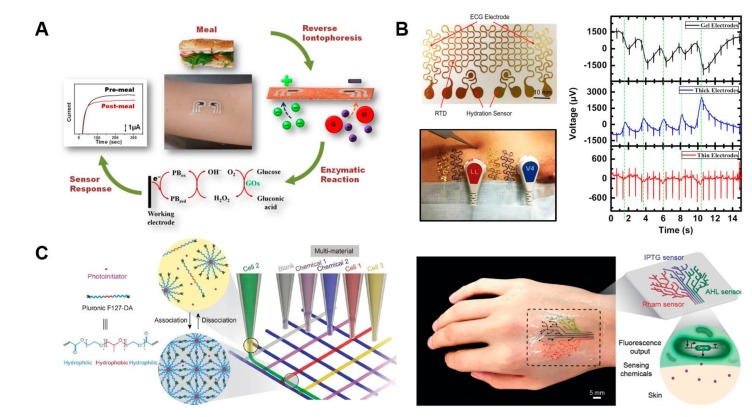
Schematic of implantable devices. (**A**) Tattoo-based glucose detection platform. (**B**) Photograph of an e-tattoo incorporating two electrocardiogram (ECG) electrodes, two hydration sensors and an RTD, all in filamentary serpentine (FS) layout. Synchronously measured ECG under skin indentation. (**C**) Schematic illustration shows direct hydrogel ink writing. The packing of Pluronic F127-DA micelles in the ink leads to a physically crosslinked hydrogel after printing; photoinitiator allows postphotocrosslinking of the living structures after printing; engineered bacterial cells are programmed to sense the signaling chemicals. 3D printed living tattoo is printed as a tree-like pattern on a thin elastomer layer and adhered to human skin. Reproduced with permission from Bandodkar, A.J.; et al. Tattoo-based noninvasive glucose monitoring: A proof-of-concept study; American Chemical Society, 2015 [98], Liu, X.; et al. 3D printing of living responsive materials and devices; Wiley, 2018 [101] and Wang, Y.; et al. Low-cost, μm-thick, tape-free electronic tattoo sensors with minimized motion and sweat artifacts; Springer Nature, 2018 [106].

**Figure 7 nanomaterials-09-00813-f007:**
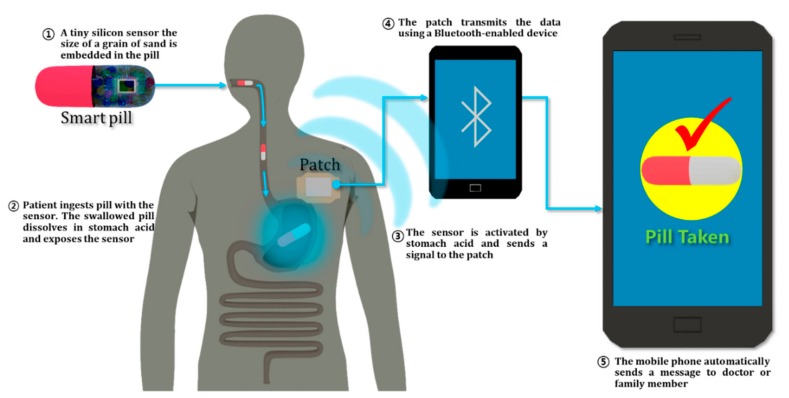
Overview of ingestible pill sensor system.

**Table 1 nanomaterials-09-00813-t001:** Summary of portable devices.

	Monitoring	Physiological & Physical Parameters (Device)	Ref.
Wrist-mounted Devices	Cardiovascular signal	heart rate, blood pulse etc. (wrist band or watch)	[20,21,22,23]
	Sweat contents	glucose, sodium etc. (wrist band or watch)	[24,25]
Head-mounted Devices	Salivary contents	lactate, uric acid and glucose etc. (mouth guard)	[32,33,34]
	Sweat contents	Lactate and potassium etc. (eyeglasses)	[30]
	Cardiovascular signal	heart rate (eyeglasses)	[29]
E-Textiles	Sweat contents	glucose and lactate etc. (textiles with electrode)	[36]
	Cardiovascular signal	heart rate and temperature etc. (leg calf)	[40,41,42]
	Physical activity	foot motion (footwear)	[43,44]
Others	Physical activity	sleep, daily activity etc. (ring, necklace and clip etc.)	[46,47,48,49]
	Physical activity	step count and sitting time (belt worn on waist)	[51,52]
	Physiological signal	ECG and direct current (belt worn on chest)	[53,54,55]

**Table 2 nanomaterials-09-00813-t002:** Comparison of advantages and disadvantages of piezoelectric, resistive and capacitive sensors.

	Advantages	Disadvantages
Piezoelectric sensor	High sensitivity High mechanical stiffness Actuation mechanism is highly resistive to environmental effect (e.g., humidity, temperature) Very fast response Broad frequency range Excellent repeatability	High impedance Small displacements Low material tensile strength Auxiliary Equipment needed Limited temperature range
Resistive sensor	Bond excellently to most surfaces Minimal sensitivity to transverse strain High frequency response High linearity Low impedance Good spatial resolution Generally unaffected by ambient conditions Ability to measure dynamic loads Simple construction	Temperature sensitive (Gage factor changes with temperature as well) compared to piezoresistive sensors strain gages have much lower sensitivity (typical gage factor 2 vs. 100 for the piezoresistive sensor)
Capacitive sensor	Be used to detect non-metallic targets Simple in construction and adjustable Detect dense targets and liquids Relatively less costly and small Higher sensitivity and can be operational with small magnitude of force Very good resolution (as low as 0.003 mm) and frequency response	Operation needs a clean environment (a capacitor is affected by temperature, humidity, pressure, dust, etc.) The measurement of capacitance is hard compare to measurement of resistance Capacitive proximity sensor are not so accurate compare to inductive sensor type

**Table 3 nanomaterials-09-00813-t003:** Summary of the attachable, implantable and ingestible devices for health monitoring.

Attachable Devices			
	Monitoring	Physiological Parameters	Ref.
Patch	Cardiovascular signal	Blood pressure and heart rate by measuring of ECG, BCG and pulse transit time with a thin, flexible patch	[59,60,61,65]
	Chemicals	Sweat volume and sweat components like hydration, glucose, lactate, pH and electrolytes	[69,71,72,73,74,77,90]
	Body temperature	Body temperature on skin	[79,82,83,84]
Contact lens	Chemicals	Glucose and lactate in tear fluid	[85,86,87,88,91]
**Implantable and Ingestible Devices**			
	**Monitoring**	**Physiological parameters**	**Ref.**
Pace-maker	Cardiovascular signal	Heartbeat for treating arrhythmias	[92,93,94,95]
Tattoo	Salivary Sweat	Monitoring respiration and pathogenic bacteria detection with tooth enamel Lactate, glucose, alcohol and electrolytes (such as ammonium) with skin worn tattoo	[96,97,98,99,100]
Bioink	Interstitial fluid	Glucose, pH and electrolytes (such as sodium)	[101,102]
Smart pill	Medicine	Medicine when drug reaches stomach with patch	[103,104,105]

**Table 4 nanomaterials-09-00813-t004:** Application of disease and examples of commercial wearable devices.

Disease	Monitoring	Product Category	Commercial Product
Metabolic disorder	Glucose Hydration	Wrist band/watch Ear appliance Patch Wrist band/watch	GlucoWatch G2 Biographer [24] Gluco Track Symphony Freestyle Libre Dexcom Patches LVL
Mosquito-borne diseases	Temperature Sweat patterns	Smart jewelry	TermoTelll bracelet
Skin disease & UV related disease	Level of UV	Smart patch Smart jewelry	MyUV Patch Netatmo JUNE
Respiratory diseases	Audio signal, heart rate, accelerations Cardiac electrical activity (ECG)	Wrist band/watch Smart patch	LG Watch Urbane W150 Moto 360 2nd Generation Savvy patch ECG sensor [116] XYZlife Patch BC1
Skeletal system diseases	Movement postural variation gait	Smart shoes	CUR Smart Pain Relief Valedo Lumo Lift
Sleep or stress related disease	Heart rate variability, Heart rate	Wrist watch/band Smart jewelry Patch	Airo Health;s anxiety tracker Oura ring [46] Motiv ring [47] Go2Sleep Kenzen Patch [117] Vital Scout
Cognitive disorder	GPS	Wrist band/watch	VegaGPSbracelet
Cardiovascular disease	Heart rate Pulse rate	Wrist watch/band	HEM serieses of OMRON
Fitness tracking	Heart rate, Calories burned, activity level Heart rate, Heart rate variability, Body temperature	Smart jewelry Ear appliance	Ear-o-smart [48] Cosinuss’ One
Others	Temperature	Patch Ear appliance	Fever scout degree

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
