# Peer review of "Evolution of Wearable Devices with Real-Time Disease Monitoring for Personalized Healthcare"

_nanomaterials, 2019, doi:10.3390/nano9060813_

Round 1
Reviewer 1 Report
> Please include some recent examples of bioresorbable devices in the implantables section
> Include micro-structured pyramid sensors for e-skin pioneered by Bao group.
> Give examples of commercialized sweat analysis devices and their use cases
> Image quality is low. Please provide better quality images.
> Please give more examples of piezoelectric devices
> Provide a comparison of different types of sensors (piezoelectric, resistive, capacitive, etc.) with a table of advantages/disadvantages
Reviewer 2 Report
In this review article, Kim et al have provided information on the recent progress and status of the wearable devices for health monitoring. Although the authors have comprehensively described the technologies, after careful reading I found that the whole content of the manuscript did not match with the title. The title says that it is about personalized therapy, however, technologies described in the manuscript are all about health monitoring by using of sensors. I did not find much information about how these technologies are related to personalized therapy. Therefore, either the title needs to be modified or the manuscript has to be rewritten. Also after searching I found a very recent review which is quite similar to this manuscript (Kim J, Campbell AS, de Ávila BE, Wang J. Wearable biosensors for healthcare monitoring, Nat Biotechnol. 2019 Apr;37(4):389-406.). This published manuscript came with more detailed description on the science of each technology. The authors have missed this reference. Therefore, although the manuscript is well written, I can not recommend to publish it in Nanomaterials.
Author Response
Please find the attachment,
Thanks
Regards,
Juyeon Jung
Response to the Reviewer’s comments:
Manuscript ID: nanomaterials-495832
Title: Evolution of wearable devices for real-time disease monitoring for personalized therapy
We appreciate your valuable comments and advice which helped us to correct errors and strengthen our manuscript.
Reviewer #2)
In this review article, Kim et al have provided information on the recent progress and status of the wearable devices for health monitoring. Although the authors have comprehensively described the technologies, after careful reading I found that the whole content of the manuscript did not match with the title. The title says that it is about personalized therapy, however, technologies described in the manuscript are all about health monitoring by using of sensors. I did not find much information about how these technologies are related to personalized therapy. Therefore, either the title needs to be modified or the manuscript has to be rewritten.
: We first modified the title to “Evolution of wearable devices with real-time monitoring for personalized healthcare” according to the reviewer’s suggestion.
Also after searching I found a very recent review which is quite similar to this manuscript (Kim J, Campbell AS, de Ávila BE, Wang J. Wearable biosensors for healthcare monitoring, Nat Biotechnol. 2019 Apr;37(4):389-406.). This published manuscript came with more detailed description on the science of each technology. The authors have missed this reference. Therefore, although the manuscript is well written, I cannot recommend to publish it in Nanomaterials.
: Researches on the development of wearable devices have become more active, and related research papers are being published rapidly.
Figure R1. Number of papers published for wearable devices during 2011-2019, analyzed by Scopus.
The review paper mentioned by reviewer are classified according to the location of application of the wearable device (epidermal-, ocular- and oral-cavity-). However, our manuscript is organized according to the evolution stage of the wearable device technology (Portable → Attachable → Implantable). Therefore, we consider this review paper to be sufficiently different and valuable. We also updated the reference and added the review paper that the reviewer mentioned [Kim, J.; Campbell, A.S.; de Ávila, B.E.; Wang, J. Wearable biosensors for healthcare monitoring. Nat. Biotechnol. 2019, 37, 389-406].
Figure R2. Development stage of wearable device (Analyst report on the wearable device industry, 2014).
We hope that this revision has satisfied all the concerns of the reviewer. We greatly appreciate your efforts in handling the manuscript.
Sincerely,
Juyeon Jung
Principal Researcher
BioNanotechnology Research Center
Korea Research Institute of Bioscience and Biotechnology
Daejeon 305-806, Republic of Korea

Reviewer 3 Report
This is a good overview of the field. I have only two comments: 1/ I feel a bit more information on smartwatches enabled monitoring would be worthy i.e. information we can get from them such as movement, PPG etc. and how this may be used for patient monitoring. 2/ Genetic circuit design and synthetic biology in the context of healthcare. I would consider talking about this subject.Author Response
Please find the attachment.
Thanks.
Regards,
Juyeon Jung

Round 2
Reviewer 2 Report
I recommend to accept this revised version.